# Antibacterial, antibiofilm and anti-quorum sensing activities of 1,2,3,5-tetrazine derivatives linked to a benzothiazole moiety

Jean Paul Dzoyem[1,2]*, Joseph Tsemeugne[3], Boniface Pone Kamdem[4,5]*,
Rostand Foyou Meupiap[3], Boris Arnaud Kuate[3], Pierre Mkounga[3],
Fabrice Fekam Boyom[4,5], Lyndy Joy McGaw[2]

1 Department of Biochemistry, Faculty of Science, University of Dschang, Dschang, Cameroon,
2 Phytomedicine Programme, Department of Paraclinical Sciences, University of Pretoria, Pretoria,
South Africa, 3 Laboratory of Natural Products and Applied Organic Synthesis (LANAPOS), Department
of Organic Chemistry, Faculty of Science, University of Yaounde I, Yaounde, Republic of Cameroon,
4 Antimicrobial and Biocontrol Agents Unit (AmBcAU), Laboratory for Phytobiochemistry and Medicinal
Plants Studies, Department of Biochemistry, Faculty of Science, University of Yaounde I, Yaounde,
Cameroon, 5 Advanced Research & Health Innovation Hub, Yaounde, Cameroon

* jean.dzoyem@univ-dschang.org, jpdzoyem@yahoo.fr (JPD); ponekamdemboniface@gmail.com (BPK)

## Abstract

A series of known tetrazine derivatives, containing benzothiazole scaffold, were prepared during the coupling reactions of selected diazotized 2-aminobenzo[d]thiazole derivatives with p-acetaminophen. The as-prepared compounds were characterized based on NMR and MS spectrometry. The antibacterial and anti-biofilm activities of the synthesized compounds were evaluated by microdilution method, whereas the anti-quorum sensing effect was carried out using assay for the inhibition of violacein formation. As a result, compounds **4a**, **4b** and **4c** revealed minimum inhibitory concentrations and minimum bactericidal concentrations ranging from 8 to 128 μg/mL and from 32 to 256 μg/mL, respectively. Compounds **4a** (52–86.5%), **4b** (57.7–79.4%) and **4c** (59.9–80.3%) prevented biofilm formation in all the four bacteria tested with percentages of inhibition more than 50%. The concentrations of **4a**, **4b** and **4c** that inhibited 50% of violacein production were found to be 62.71, 28.56 and 107.45 μg/mL, respectively, thus attesting that these compounds possess anti-quorum sensing activity. Noteworthy, our previous investigation attested that these compounds are non-cytotoxic on the human mammalian cells Vero. This novel contribution demonstrates the antibacterial, antibiofilm and anti-quorum sensing activities of tetrazine-based benzothiazoles, which might be prospected as scaffolds for the discovery of efficient antibiotics with decreased risk of microbial drug resistance.

## Introduction

Resistance to currently available antibiotics has become a grave menace to the treatment of infectious diseases [1,2]. According to the 2019's World Health Organization

**Data availability statement:** All relevant data are within the manuscript and its Supporting Information files.

**Funding:** The author(s) received no specific funding for this work.

**Competing interests:** The authors have declared that no competing interests exist.

(WHO) report, antibacterial resistance was accountable for 1.27 million deaths worldwide [3,4]. The main mechanisms of this notorious phenomenon include enzymatic alteration and efflux pump systemic elimination of antimicrobials, structural modification of bacterial target proteins, changes in bacterial membrane permeability, etc. In the meantime, many pathogenic bacteria can form a dense biofilm [5,6], thus making bacteria highly resistant to antibiotics [7]. In fact, the contribution of biofilms in antimicrobial resistance is highly complex and may significantly drive resistance [8–10]. By controlling the formation of biofilms and drug efflux pumps, the quorum sensing system plays a crucial role in developing bacterial drug-resistant pathways [11]. Growing evidence has demonstrated the link between quorum sensing and biofilm development [12–14]. As a matter of fact, a quorum sensing is a density-dependent cell-signalling mechanism by which bacteria crosstalk to each other [14,15]. Because of its connection to bacterial pathogenicity, virulence and biofilm formation, quorum sensing has gained more research attention in the last decade [16,17]. Thus, targeting quorum sensing and biofilm formation would be a prominent approach to unravel antibiotic resistance in pathogenic bacteria [18,19].

Modern research has substantially identified antibacterial compounds with anti-quorum sensing and anti-biofilm properties [6,20–24], even though almost no such chemotype is reported to have succeeded the last clinical trial stage in humans.

Thus, the development of effective chemotypes that inhibit bacterial growth while preventing biofilm formation and attenuating quorum sensing-dependent virulence factors is of paramount importance.

Benzothiazole is a heterocyclic and bicyclic pharmacophore that contains benzene fused with 1,3-thiazole skeleton [25,26]. A number of scientists have established the potential of benzothiazole and its derivatives as antimicrobial hit compounds [27–29].

On the other hand, modern research on tetrazine derivatives in relation to their antibacterial activity has been reported by many researchers [30,31].

The incorporation of 1,2,3,5-tetrazine derivatives into the benzothiazole structure introduces new chemical functionalities that can potentially disrupt bacterial biofilms and interfere with quorum sensing mechanisms. This dual approach aims not only to inhibit bacterial growth but also to prevent biofilm formation and attenuate virulence factors, thereby offering a comprehensive strategy to combat bacterial infections.

In our previous research investigation, we demonstrated the cytotoxic effects of 1,2,3,5-tetrazine tethered benzothiazole derivatives against a number of cancer cells, including A549, Hela and MCF-7 cells. However; these compounds were non cytotoxic vis-à-vis the human mammalian cells Vero.

In our continuing effort to search for effective antibacterial hit compounds that might aid in drug discovery campaigns, this study sought to investigate the inhibitory effects of certain 1,2,3,5-tetrazine tethered benzothiazole derivatives on the growth of selected bacteria. Moreover, anti-biofilm and anti-quorum sensing activities of these compounds are also investigated.

## Materials and methods

### Chemistry

**General.** The reagents of analytical grade were purchased from commercial sources and used without any further purification. [1]HNMR spectra were measured with a 400 MHz spectrometer NMR Bruker Advance 400 at room temperature in DMSO-$d_6$ with tetramethylsilane as the internal reference. [13]C-NMR spectra were recorded in DMSO-$d_6$ with a 100 MHz spectrometer NMR Bruker Advance 400. UV-visible absorption spectra were recorded on Beckman U-640 Spectrophotometer, using samples' solutions of concentration $5 \times 10^{-5}$ mol.L$^{-1}$. Infrared spectra were taken in KBr on a Perkin Elmer FT-IR 2000 spectrophotometer. Masss spectra were measured with a Waters Xevo TQD tandem quadruple mass spectrometry system running in MS scan mode, 1 minute of acquired spectra were combined and centroided. Melting points were obtained with a Buchii melting point apparatus and are uncorrected. The Thin Layer Chromatography (T.L.C.) was carried out on Eastman Chromatogram Silica Gel Sheets (13181; 6060) with fluorescent indicators. A mixture of hexane and ethyl acetate (4:6) was used as the eluent and iodine was used for the visualization of the chromatograms.

**Preparation of diazonium salt solution.** As per a previously reported protocol [32], dried sodium nitrite (0.69 g, 10 mmol) was slowly added over a period of 30 minutes to concentrated sulphuric acid (10 mL) with occasional stirring. The solution was cooled to 0–5 °C. Compound **1** was dissolved in DMSO (10 mL) and cooled to 0–5 °C. The nitrosyl sulphuric acid solution was added to the solution of **1** and the temperature was maintained between 0–5 °C. The clear diazonium salt solution thus obtained consisting of the *in situ*-formed intermediate **2**, was used immediately in the coupling reactions.

**General procedure for the preparation of the coupling products (4).** Acetaminophen (**3**) (1.51 g, 10 mmol) or 2-amino-6-nitrobenzothiazole (**1b**) (1.952 g, 10 mmol) was dissolved in DMSO (10 mL) and then cooled in an ice-bath at 0–5 °C. A prepared diazonium solution of **2** was added drop wise over 1 hour, and then 15 mL of sodium acetate solution (10%) was added to the mixture. The pH of the mixtures was in the range 9–11. The solid precipitate was collected on a filter and crystallised from methanol to give the title compound.

**N-(3-((5,6-dimethylbenzo[d]thiazol-2-yl)diazenyl)-4-hydroxyphenyl)acetamide (4a).** Compound **4a** was obtained in 58% yield as red powder; m.p. 118–120 °C; [Litt: 119–121 °C, [32]; [1]H-NMR (DMSO-$d_6$, 400MHz): δ 10.67 (s, 1H, O-H), 9.94 (s, 1H, N-H), 8.10 (d, 1H, J = 4.0 Hz, H-2'), 7.89 (s, 1H, H-4), 7.84 (s, 1H, H-7), 7.60 (dd, 1H, J = 4.0 and 8.0 Hz, H-6'), 7.06 (d, 1H, J = 8.0 Hz H-5'), 2.36 (s, 6H, 2CH$_3$), 2.01 (s, 3H, COCH$_3$); [13]C-NMR (DMSO-$d_6$, 100 MHz): δ 174.6 (CO), 168.1 (C-2), 153.6 (C-3a), 131.3 (C-4a), 124.2 (C-4), 137.4 (C-5), 132.1 (C-6), 122.5 (C-7), 138.5 (C-1'), 108.1 (C-2'), 136.1 (C-3'), 151.1 (C-4'), 118.8 (C-5'), 128.1 (C-6'), 23.8 (CH$_3$CO), 19.9 (CH$_3$), 19.6 (CH$_3$); UV-Vis λ$_{max}$ (DMSO) (Log ε): 274 (5.00), 327 (4.39), 364 (4.49), 452 (4.01) nm; IR (KBr) υ$_{max}$: 3248 (O-H and N-H), 1659 (C=O), 1604−1557 (C=C), 1483−1450 (N=N), 1274 (C-S), 1239 (C-S), 861−510 (Ar def C=N str thiazole) cm$^{-1}$. (ESI+ ) *m/z* (%) 394 (8), 389 (10), 375 (13), 372 (7), 316 (65), 304 (48), 283 (19), 202 (14), 192 (21), 150 (70); Anal. Calcd. for C$_{17}$H$_{22}$N$_4$O$_5$S: C, 59.98; H, 4.74; N, 16.46; S, 9.42. Found: C, 59.63; H, 4.80; N, 16.41; S, 9.40. Rf = 0.62.

**4-((5-acetamido-2-hydroxyphenyl)diazenyl)-3-(2-mercapto-4,5-dimethylphenyl)-7,8-dimethylbenzo[4,5]thiazolo [2,3-d][1,2,3,5]tetrazine-3,5-diium sulfate (4a').** Compound **4a'** was obtained in 34% yield as brown powder; m.p. 318–319 °C; [Litt. 318–320 °C [32]; [1]H-NMR (DMSO-*d6*, 400 MHz): δ 11.35 (s, 1H, OH), 10.67 (s, 1H, NH), 8.37 (s, 1H, H-9'''), 8.13 (d, 1H, J = 2.8 Hz, H-6'), 7.93 (s, 1H, H-6''), 7.77 (s, 1H, H-6''), 7.69 (s, 1H, H-3''), 7.63 (dd, 1H, J = 8.8 and 2.8 Hz, H-4'), 7.09 (d, 1H, J = 8.8 Hz, H-3'), 2.50, 2.39, 2.37, 1.23 (s, 12H, CH$_3$), 2.04 (s, 3H, COCH$_3$); [13]C-NMR (DMSO-*d6*, 100 MHz): δ 198.6 (C-4), 174.5 (C=O), 168.1 (C-6), 153.5 (C-1''), 151.0 (C-2'), 138.4 (C-5'''), 137.4 (C-8''' and C-5'), 136.0 (C-5''), 132.1 (C-7'''), 131.3 (C-4''' and C-2''), 128.1 (C-1'), 124.2 (C-4''), 122.4 (C-3''), 122.1 (C-9'''), 121.6 (C-6'''), 118.8 (C-6''), 118.3 (C-4'), 108.0 (C-3'), 105.4 (C-6'), 23.8 (COCH$_3$), 23.7, 19.86, 19.7, 19.6 (Ph-CH$_3$); UV-Vis λ$_{max}$ (MeOH) (Log ε): 227 (4.06), 257 (4.12), 272 (4.26), 290 (4.09), 295 (4.08), 302 (4.12), 325 (4.19), 348 (4.18), 355 (4.19), 399 (4.23), 445 (4.25), 486 (4.22) nm; IR (KBr) ν$_{max}$: 3887−3282 (O-H and N-H), 2920 (ArC-H), 2324 (S-H), 1664−1655 (C=O), 1533 (C=C), 1490−1449 (N=N), 1370 (δ$_{tetrazine\ ring}$), 1269 (C-S), 1240 (C-O), 889 (δ$_{tetrazine\ ring}$) cm$^{-1}$; ms: (ESI+ ) *m/z* (%) 699 (8),

673 (9), 643 (11), 659 (75), 601 (10), 599 (58), 581 (22), 485 (41), 410 (34), 409 (74), 316 (100), 166 (47); Anal. Calcd. for $C_{26}H_{33}N_7O_{10}S_3$: C, 44.63; H, 4.75; N, 14.01; S, 13.74. Found: C, 44.59; H, 4.80; N, 14.05; S, 13.71. Rf = 0.30.

**N-4-Hydroxy-2,3-bis[3-(3-Nitro-benzenethiol-5)-yl-7-nitro-9-thia-1,2-diaza-3,4a-diazonia-fluorene-4)-yl-diazenyl]-5,6-bis[(6-Nitro-benzothiazol-2)-yl-diazenyl]-phenyl-acetamide disulfate (4b).** Compound **4b** was obtained in 67% yield as brown powder; m.p. 169–172 °C; [Litt. 171–173 °C [32]]; $^1$H-NMR (DMSO-$d_6$, 400 MHz): δ 8.65 (d, 1H, J = 2.4 Hz, H-8$^v$), 8.58 (d, 1H, J = 2.8 Hz, H-4'), 8.43 (dd, 1H, J = 8.8 and 2.8 Hz, H-5'), 8.43 (dd, 1H, J = 9.2 and 2.4 Hz, H-6$^v$), 8.38 (dd, 1H, J = 8.8 and 2.4 Hz, H-4$^{iv}$), 8.38 (dd, 1H, J = 9.2 and 2.0 Hz, H-6$^{vii}$), 8.31 (d, 1H, J = 9.2 Hz, H-5$^v$), 8.20 (dd, 1H, J = 6.4 and 2.0 Hz, H-4$^{viii}$), 8.19 (d, 1H, J = 2.4 Hz, H-2$^{iv}$), 8.18 (d, 1H, J = 6.4 Hz, H-2$^{viii}$), 8.10 (dd, 1H, J = 8.8 and 2.4 Hz, H-5"), 7.85 (d, 1H, J = 8.8 Hz, H-4"), 7.68 (d, 1H, J = 2.0 Hz, H-5$^{viii}$), 7.42 (d, 1H, J = 9.2 Hz, H-8$^{vii}$), 7.35 (d, 1H, J = 2.0 Hz, H-5$^{vii}$), 7.28 (d, 1H, J = 8.8 Hz, H-7'), 7.13 (d, 1H, J = 2.4 Hz, H-7"), 7.05 (d, 1H, J = 8.8 Hz, H-5$^{iv}$), 3.17 (s, 2H, SH), 2.05 (s, 3H, COCH$_3$); $^{13}$C-NMR (DMSO-$d_6$, 100 MHz): δ 180.4 (CO), 171.7 (C-4'''), 170.5 (C-4$^{vi}$), 169.8 (C-2'), 168.3 (C-2"), 168.0 (C-4), 158.2 (C-3$^{vii}$), 155.8 (C-3a'), 155.3 (C-3a"), 155.1 (C-3$^{iv}$), 152.5 (C-7$^v$), 145.3 (C-6$^{iv}$), 143.7 (C-6$^{vii}$), 143.2 (C-7$^{viii}$), 142.4 (C-6'), 141.9 (C-6"), 140.7 (C-9a''' and 9a$^{vi}$), 138.6 (C-1$^{vii}$), 135.6 (C-7a"), 134.4 (C-7a'), 132.2 (C-5a$^v$ and 5a$^{viii}$), 131.7 (C-1$^{iv}$), 131.4 (C-1), 129.9 (C-8a$^{viii}$), 125.0 (C-8a$^v$), 124.5 (C-6$^{viii}$), 124.4 (C-6$^v$), 122.7 (C-5$^{iv}$), 122.3 (C-5$^{vii}$), 122.1 (C-5$^v$), 121.9 (C-5$^{viii}$), 121.8 (C-8$^{viii}$), 121.8 (C-4$^{iv}$), 121.5 (C-8$^v$), 120.4 (C-4$^{vii}$), 119.8 (C-4'), 119.1 (C-4"), 119.0 (C-2), 119.0 (C-2$^{vii}$), 118.6 (C-6), 118.4 (C-5'), 118.2 (C-5"), 117.6 (C-2$^{iv}$), 116.8 (C-7"), 116.7 (C-7'), 111.4 (C-5), 107.1 (C-3), 23.8 (CO$\underline{C}$H3); UV-Vis $\lambda_{max}$ (MeOH) (Log ε): 272 (4.67), 348 (4.87), 437 (4.25), 483 (4.25), 555 (3.75) nm; IR (KBr) ν$_{max}$: 3285 (O-H and N-H), 3097 (ArC-H), 1654 (C=O), 1599 (C=N), 1512 (C=C), 1442 (N=N), 1269 (C-S), 1234 (C-O), 910−502 (Ar def C=N str thiazole) cm$^{-1}$; ms: (ESI$^+$) m/z (%) 1052 (4), 996 (3), 882 (3), 713 (3), 694 (3), 659 (3), 637 (7), 599 (13), 409 (16), 317 (26), 316 (81); Anal. Calcd. for $C_{50}H_{33}N_{25}O_{26}S8$: C, 36.26; H, 2.01; N, 21.14; S, 15.48. Found: C, 36.24; H, 1.98; N, 21.17; S, 15.43. Rf = 0.53.

**3,11-dinitrobenzo[4,5]thiazolo[3,2-c]benzo[4,5]thiazolo[3,2-e][1,2,3,5]tetrazine-8,14-diium sulfate (4c).** Compound **4c** was obtained in 41% yield as orange powder; m.p. 244–246 °C; [Litt. 245–247 °C [32]]; $^1$H-NMR (DMSO-$d_6$, 600 MHz): δ 8.95 (d, 1H, J = 1.8 Hz, H-7), 8.69 (d, 1H, J = 2.4 Hz, H-7'), 8.29 (s, 2H, NH), 8.25 (dd, 1H, J = 2.8 and 8.8 Hz, H-5), 8.11 (dd, 1H, J = 2.4 and 8.8 Hz, H-5'), 7.90 (d, 1H, J = 8.8 Hz, H-4), 7.42 (d, 1H, J = 8.8 Hz, H-4'); $^{13}$C-NMR (DMSO-$d_6$, 150 MHz): δ 153.2 (C-2), 155.0 (C-3a), 121.0 (C-4), 122.5 (C-5 and C-5'), 144.0 (C-6), 119.3 (C-7), 132.6 (C-7a), 172.3 (C-2'), 158.7 (C-3a'), 117.3 (C-4'), 141.2 (C-6'), 118.3 (C-7'), 131.9 (C-7a'); **UV-Vis (MeOH)** $\lambda_{max}$ (log ε): 260 (4.54), 285 (4.48), 352 (4.66), 393 (4.62), 421 (4.63), 450 (4.64), 472 (4.62); **IR (KBr)** υ$_{max}$/cm$^{-1}$: 3097 (C$_{Ar}$-H), 1556 (C=N), 1444 (N=N), 1336 (C$_{Ar}$-NO$_2$), 1120 (C-S), 1514 (C=C); ms: (ESI$^+$) m/z (%) 590 (5), 568 (3), 562 (10), 558 (15), 554 (10), 550 (17), 514 (35), 450 (76), 378 (10), 294 (13), 248 (34). Anal. Calcd. for: $C_{14}H_{18}N_6O_{14}S_3$: C, 28.48; H, 3.07; N, 14.23; S, 16.29. Found: C, 28.50; H, 3.1; N, 14.25; S, 16.32. Rf = 0.45.

**1,2-bis(6-nitrobenzothiazol-2-yl)diazene-1,2-diium sulfate (4c').** Compound **4c'** was obtained in 28% yield as red powder; m.p. 243–245 °C; [Litt. 243–245 °C [32]]; $^1$H-NMR (DMSO-$d_6$, 600 MHz): δ 8.69 (d, 2H, J = 2.4 Hz, H-7 and H-7'), 8.24 (2H, s, NH), 8.10 (dd, 2H, J = 2.4 and 8.8 Hz, H-5 and H-5'), 7.42 (d, 1H, J = 8.8 Hz, H-4 and H-4'); $^{13}$C-NMR (DMSO-$d_6$, 150 MHz): δ 172.3 (C-2 and C-2'), 159.1 (C-3a and C-3a'), 117.3 (C-4 and C-4'), 122.5 (C-5 and C-5'), 141.2 (C-6 and C-6'), 118.2 (C-7 and C-7'), 132.1 (C-7a and C-7a'). **UV-Vis (MeOH)** $\lambda_{max}$ (log ε): 269.7 (4.53), 279.6 (4.30), 351.3 (5.17); **IR (KBr)** υ$_{max}$/cm$^{-1}$: 3508 (N-H), 3068 (C$_{Ar}$-H), 1644 (C=N), 1568 (C=C), 1486 (N=N), 1282 (C$_{Ar}$-NO$_2$), 1120 (C-S); ms: (ESI$^+$) m/z (%) 484 (5), 452 (100), 456 (5), 466 (8), 438 (10), 428 (17). Anal. Calcd. for $C_{14}H_8N_6O_8S3$: C, 34.71; H, 1.66; N, 17.35; S, 19.85 Found: C, 34.73; H, 1.70; N, 17.33; S, 19.88. Rf = 0.48.

## Antimicrobial susceptibility and antibiofilm assays

**Microbial strains.** Strains of *Klebsiella aerogenes* ATCC 130148, *Acinetobacter baumannii* ATCC BAA-1605, *Enterococcus faecium* ATCC700221, *Staphylococcus epidermidis* ATCC35984 and *Chromobacterium violaceum* ATCC12472 from the American Type Culture Collection (ATCC) were used. They were maintained in Muller Hinton agar (MHA) at 37°C, while *Chromobacterium violaceum* strain was maintained in Luria–Bertani (LB) agar at 25°C.

**Determination of the minimum inhibitory concentration (MIC) and the minimum bactericidal concentration (MBC).** MIC and MBC values were determined by the broth microdilution method using Muller Hinton broth (MHB). Stock solutions of compounds and reference antibacterial gentamicin were prepared in 100% dimethyl sulfoxide (DMSO; Sigma), and twofold serial dilutions were prepared in media in amounts of 100 µL per well in a 96-well plate. Then, 100 µL of a bacterial suspension was added to each well of the plate except those of the sterility control, resulting in a final inoculum of $1.5 \times 10^6$ CFU/mL. The final concentration of samples ranged 0.125 to 256 µg/mL. The final concentration of DMSO was lower than 2.5% and does not affect the bacterial growth. The medium without the agents was used as a growth control, and the blank control contained only the medium. Gentamicin (final concentrations' range: 0.031–64 µg/mL) was used as a positive control. The microtitre plates were incubated at 37°C for 24 h. The assay was repeated three times in triplicate. The MIC of the samples was detected following the addition (40 µL) of 0.2 mg/mL *p*-iodonitrotetrazolium chloride and incubation at 37°C for 30 minutes. Viable microorganisms reduced the yellow dye to a pink color. MIC was defined as the lowest sample concentration that prevented this change and exhibited complete inhibition of bacterial growth.

The MBC was determined by adding 50 µL of the suspensions from the wells, which did not show any growth after incubation during MIC assays, to 150 µL of fresh MHB. These suspensions were incubated at 37°C for 48 h. The MBC was determined as the lowest concentration of sample that inhibited bacterial viability.

**Determination of the minimum biofilm inhibitory concentration (MBIC$_{50}$) and the minimum biofilm eradicating concentration (MBEC$_{50}$).** In a preliminary experiment, all the test compounds were evaluated at their MIC concentration, and those with more than 50% biofilm inhibition/eradication were selected for a dose–response assay to determine the MBIC$_{50}$ and MBEC$_{50}$ values. This test was carried out by the broth microdilution method as previously described [33]. Briefly, 100 µL of MHB supplemented with 2% glucose containing the samples was introduced into the first wells followed by a serial twofold dilution. Subsequently, 100 µL of bacterial suspension was added to all wells except those of the sterility control, resulting in a final inoculum of $1.5 \times 10^6$ CFU/mL, followed by incubation at 37°C for 24 h. After incubation, the plate was washed three times with phosphate-buffered saline (PBS; pH 7.2) to remove non-adherent bacteria cells. Wells containing MHB without bacteria served as the negative control. The remaining bacterial cells that attached to the well surface were considered as true biofilm. Then, the plates were stained with 0.1% crystal violet solution for 20 min at room temperature. After staining, the plates were washed three times with PBS. Then, the plates were air-dried and destained with 150 µL of 95% ethanol (v/v) for 30 min. Finally, the optical density was measured at 590 nm using a microplate reader (BioTek Epoch Microplate Spectrophotometer). Untreated wells and wells containing broth only were used as positive and blank controls, respectively, and the percentage of biofilm inhibition was calculated by using the following formula:

$$\% \text{ inhibition} = 100 - [(OD_{sample} - OD_{blank})/(OD_{control} - OD_{blank}) \times 100]$$

The MBEC$_{50}$ was determined under the same conditions as the MBIC$_{50}$, with the only difference being that the biofilm was allowed to form for 24 h before treatment with the samples.

Median MBIC$_{50}$ and MBEC$_{50}$ values were defined as the concentration inhibiting 50% of biofilm formation and pre-formed biofilm, respectively. This was calculated by plotting the percentage of inhibition or eradication versus the concentrations using GraphPad Prism software. Samples were tested in triplicate, and experiments were repeated three times.

## Anti-quorum sensing assay

**Inhibition of violacein production.** The inhibition of violacein production was performed according to a previously described method [34] and miniaturized in a 48-well microplate. This was achieved by transferring 1000 µL of test compounds' solutions into the first well of a 48-well microplate, followed by a serial twofold dilution in LB broth. Then, 500 µL of *C.*

*violaceum* inoculum, standardized at $3 \times 10^6$ CFU/mL, was added to all wells except those of the sterility control to obtain a final concentration of 4–256 μg/mL. Vanillin, at final concentrations ranging from 0.5 to 1024 μg/mL, was used as a reference compound [35]. Plates were properly sealed with parafilm and incubated in an orbital shaker (140 rpm) at 30°C for 24 h. The MIC was defined as the minimum concentration inhibiting visible bacterial growth and therefore preventing the production of purple pigmentation. The minimum quorum sensing inhibitory concentration (MQSIC) was defined as the lowest compound's concentration allowing bacterial growth (shown by turbidity) without the visible production of purple pigmentation.

**Quantification of violacein.** The inhibitory effect of selected compounds (**4a**, **4b** and **4c**) on violacein production was further quantified using a spectrophotometric method. After collection of MIC and MQSIC data, the plates were centrifuged at 4000 rpm for 20 min, and the supernatant was discarded. Then, the bacterial pellet was resuspended in 1 mL of DMSO, and the plates were further left in an orbital shaker for 10–15 min. Then, 200 μL of the supernatant was transferred into a 96-well microplate, and the optical density was measured at 595 nm. The percentage of violacein inhibition was calculated using the following formula:

$$\% \text{ violacein inhibition } = 100 - [(OD_{sample} - OD_{blank})/(OD_{control} - OD_{blamk}) \times 100].$$

The $IC_{50}$ values of the test compounds were defined as the concentrations inhibiting 50% of violacein production. These values were calculated by plotting the percentages of violacein production versus the concentrations using GraphPad Prism software. Samples were tested in triplicate, and each experiment was repeated three times.

### Kinetics of bacterial growth and biofilm formation at sub-MIC concentrations

The effect of sub-MIC concentrations on bacterial growth and biofilm was evaluated by performing the kinetics of bacterial growth and biofilm formation at MIC, 1/2 xMIC, 1/4 xMIC, 1/8 xMIC, 1/16 xMIC and 1/32xMIC. Then, the average of optical density values obtained at 570 nm, were plotted against the concentrations.

### Statistical analysis

The data are presented as the mean ± standard deviation (SD) of three independent experiments. Statistical differences between the $IC_{50}$ values inhibiting the violacein production of samples and the reference compound (vanillin) were assessed by two-way ANOVA followed by Sidak's multiple comparisons test in GraphPad Prism software.

## Results

### Chemistry

The synthesis of 1,2,3,5-tetrazine and azo dyes **4** is shown in Fig 1. All synthesized compounds were synthesised according to previous experimental procedure [32]. The yields, the melting points and all the spectroscopic data for these compounds described in the present study are in full agreement with those originally reported [32].

### Assays for the inhibition of bacteria

Table 1 summarizes the minimum inhibitory concentrations (MICs) and minimum bactericidal concentrations (MBCs) of 1,2,3,5-tetrazine derivatives upon screening against *Klebsiella aerogenes*, *Acinetobacter baumannii*, *Enterococcus faecium* and *Staphylococcus epidermidis*. The incubation of test compounds with the four pathogens afforded MIC and MBC values ranging from 8 to 256 μg/mL and from 64 to 256 μg/mL, respectively (Table 1). Against *Klebsiella aerogenes*, *Acinetobacter baumannii*, and *Enterococcus faecium*, the most promising compound *viz.* **4b** showed bactericidal trend as evidenced by the ratios MBC/MIC (128/64, 64/16 and 32/8 = **4**), which is more than 2. Also, compound **4a** (MBC/MIC: 128/16 = **8**) and **4c** (MBC/MIC: 64/16 = **4**) were found to be bactericidal when incubated with *Staphylococcus epidermidis*. Gentamicin, the standard antibiotic agent showed antibacterial activity against the tested bacteria with MIC values ranging from 0.25 to 2 μg/mL.

**Fig 1. Reactions' sequences to compounds 4.**

## Anti-biofilm activity

The anti-biofilm effect of 1,2,3,5-tetrazine derivatives was evaluated on biofilms formed by *Klebsiella aerogenes*, *Acineto-bacter baumannii*, *Enterococcus faecium*, and *Staphylococcus epidermidis*. The percentage of biofilm formation and eradication are summarized in Table 2. The degree of anti-biofilm activity of the test compounds was classified as highly and poorly actives for percentages of inhibition of >50%, and $0 < \% < 50$, respectively. Any compound with inhibition percentage

**Table 1. The minimum inhibitory concentrations (µg/mL) and the minimum bactericidal concentrations (µg/mL) of synthesized 1,2,3, 5-tetrazine derivatives.**

| Samples | Bacterial strains | | | | | | | |
|---|---|---|---|---|---|---|---|---|
| | Ka | | Ab | | Ef | | Se | |
| | MIC | MBC | MIC | MBC | MIC | MBC | MIC | MBC |
| 1a | – | – | – | – | 256 | 256 | 256 | – |
| 1b | 256 | – | 256 | – | 128 | 256 | 128 | 256 |
| 4a | 128 | 128 | **64** | 128 | **32** | 64 | **16** | 128 |
| 4a' | 256 | – | – | – | 128 | 256 | 128 | 256 |
| 4b | **64** | 128 | **16** | 64 | **8** | 32 | **32** | 64 |
| 4c | 128 | 256 | **32** | 128 | **32** | 128 | **16** | 64 |
| 4c' | 256 | – | 128 | – | – | – | 256 | – |
| Gentamicin | 2 | 4 | 1 | 2 | 0.5 | 2 | 0.25 | 1 |

– =>256 µg/mL *Ka: Klebsiella aerogenes, Ab: Acinetobacter baumannii, Ef: Enterococcus faecium, Se: Staphylococcus epidermidis.*

of antibiofilm formation of 0 was considered inactive [36]. Compounds **4a** (52–86.5%), **4b** (57.7–79.4%) and **4c** (59.9–80.3%) prevented biofilm formation in all the bacteria tested with percentages of inhibition >50%.

Similarly, compounds **4b** (55.9–61.0%) and **4c** (53.0–65.3%) eradicated biofilms formed by *K. aerogenes*, *A. baumannii* and *S. epidermidis* with percentages of eradication of >50%. Moreover, compound **4a** eradicated biofilms formed by *K. aerogenes* and *A. baumannii* with percentages of inhibition of 52.4 and 75.4%, respectively. Gentamicin, the antibiotic that was used as a positive control showed percentages of inhibition for biofilm formation and eradication ranging from 95.7 and 99.4% and from 72.2 and 86.2%, respectively (Table 2).

Table 3 summarizes the minimum biofilm inhibitory concentrations (MBIC$_{50}$s) and minimum biofilm eradicating concentrations (MBEC$_{50}$s) of compounds (**4a**, **4b** and **4c**) that exhibited more than 50% inhibition upon anti-biofilm formation and eradication assays. The minimum biofilm inhibitory concentrations (MBIC$_{50}$s) ranged from 5.29 to 54.91 µg/mL, 7.79 to 45.23 µg/mL, and 8.56 to 87.35 µg/mL for compounds **4a**, **4b** and **4c**, respectively (Table 3). Moreover, compounds **4a**, **4b** and **4c** displayed minimum biofilm eradication concentrations (MBIC$_{50}$s) of 98.62 and 60.98 µg/mL, 54.21 and 15.70 µg/mL, as well as 105.08 and 29.55 µg/mL, respectively, when tested against *Klebsiella aerogenes* and *Acinetobacter baumannii* (Table 3). All three compounds showed minimal growth inhibition at sub-MIC concentrations (≤64 µg/mL), with optical density (OD) curves closely paralleling the untreated control. Notably, **4b** demonstrated the most pronounced separation between biofilm inhibition and growth curves, maintaining >80% biofilm reduction while showing <10% growth impact at 32 µg/mL (Supporting data S1 Fig). Similarly, the three compounds showed concentration-dependent suppression of violacein production, with **4b** exhibiting the steepest dose-response curve, achieving 50% inhibition at as low as 28.56 µg/mL concentration while permitting 90% of *Chromobacterium violaceum* growth (Supporting data S2 Fig).

## Assay for the inhibition of quorum sensing

**Inhibition of violacein formation.** An indicator strain of bacteria, i.e., *Chromobacterium violaceum* 12472 was used to test the inhibitory potential of the 1,2,3,5-tetrazine derivatives (**4a**, **4b** and **4c**) on the production of violacein, a water soluble pigment that result from the disruption of quorum-sensing signals or inhibition of cell growth in a number of gram negative bacteria [37,38]. Herein, the inhibitory effects of compounds **4a**, **4b** and **4c** vis-à-vis violacein formation was evaluated at concentrations ranging from 0 to 256 µg/mL and the results are illustrated in Fig 2. As much as 256 µg/mL of compounds **4a**, **4b** and **4c** completely inhibited violacein production as evidenced by the trends of the curves that almost overlapped with the x axis at 256 µg/mL concentration. 50% inhibition of violacein formation was observed at approximately 32, 64 and 110 µg/mL concentrations for compounds **4a**, **4b** and **4c**, respectively, vs vanillin (50% inhibition

**Table 2. Percentages of inhibition of biofilm formation and eradication of synthesized 1,2,3,5-tetrazine derivatives.**

| Samples | Biofilm formation inhibition (%) | | | | Biofilm eradication (%) | | | |
|---|---|---|---|---|---|---|---|---|
| | *Ka* | *Ab* | *Ef* | *Se* | *Ka* | *Ab* | *Ef* | *Se* |
| **1a** | 4.8±0.5 | 8.5±1.2 | 18.9±0.6 | 15.7±1.7 | −6.4±1.3 | 3.5±0.2 | 3.5±0.1 | 12.5±1.5 |
| **1b** | 13.7±0.9 | 15.7±1.4 | 25.3±0.4 | 35.2±2.7 | 5.6±0.6 | 5.0±1.2 | 5.4±0.3 | 24.3±2.7 |
| **4a** | 68.5±5.2 | 86.5±5.2 | 56.6±3.4 | 52.0±3.5 | 52.4±4.6 | 75.4±4.7 | 34.5±2.4 | 42.0±4.3 |
| **4a'** | 28.9±3.4 | 4.3±0.4 | 37.4±3.8 | 28.1±1.6 | 13.5±1.8 | 4.4±1.0 | 0.09±0.0 | 20.1±2.6 |
| **4b** | 65.8±6.9 | 79.4±5.8 | 57.7±2.4 | 70.3±4.6 | 55.9±3.6 | 66.3±5.2 | 45.8±2.4 | 61.0±4.2 |
| **4c** | 65.8±5.0 | 80.3±8.7 | 59.9±4.1 | 63.0±4.2 | 59.0±4.8 | 65.3±5.7 | 48.0±3.5 | 53.0±5.8 |
| **4c'** | 32.0±2.5 | 22.5±2.6 | 9.0±0.2 | 21.6±1.8 | 15.9±1.4 | 10.0±2.1 | −5.4±0.7 | 19.0±1.2 |
| **Gentamicin** | 95.7±6.2 | 99.1±7.9 | 98.0±8.0 | 99.4±5.1 | 84.7±6.5 | 75.2±6.4 | 72.2±8.5 | 86.2±8.5 |

*Ka* *Klebsiella aerogenes*, **Ab:** *Acinetobacter baumannii* **Ef:** *Enterococcus faecium*, **Se:** *Staphylococcus epidermidis*.

**Table 3. MBIC50 and MBEC50 values (µg/mL) of synthesized 1,2,3,5-tetrazine derivatives against bacteria pathogen strains.**

| Samples | $MBIC_{50}$ (µg/mL) | | | |
|---|---|---|---|---|
| | *Ka* | *Ab* | *Ef* | *Se* |
| **4a** | 54.91±5.3 | 44.34±4.1 | 31.87±2.8 | 5.29±1.1 |
| **4b** | 45.23±4.7 | 6.54±1.0 | 7.79±1.5 | 19.22±4.8 |
| **4c** | 87.35±7.7 | 18.33±2.5 | 30.51±3.1 | 8.56±1.3 |
| **Gentamicin** | 1.22±0.1 | 2.28±0.4 | 1.03±0.2 | 0.11±0.0 |
| | $MBEC_{50}$ (µg/mL) | | | |
| **4a** | 98.62±8.5 | 60.98±5.4 | nd | nd |
| **4b** | 54.21±4.9 | 15.70±1.9 | nd | 25.44±2.2 |
| **4c** | 105.08±9.4 | 29.55±3.1 | nd | 10.45±2.1 |
| **Gentamicin** | 1.92±0.1 | 0.95±0.0 | 1.75±0.2 | 0.18±0.0 |

- = >1024 µg/mL, Ka: *Klebsiella aerogenes*, Ab: *Acinetobacter baumannii* Ef: *Enterococcus faecium*, Se: *Staphylococcus epidermidis*.

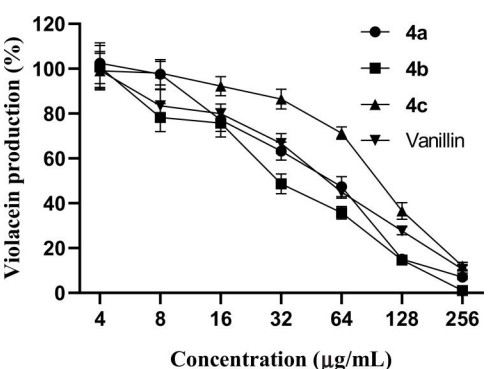

**Fig 2. Percentage of violacein production in *Chromobacterium violaceum* by the most active synthesized 1,2,3,5-tetrazine derivatives (4a, 4b and 4c).**

**Table 4. Minimum quorum sensing inhibitory concentrations (MQSICs), minimum inhibitory concentrations (MICs), and concentrations inhibiting 50% of violacein production (IC50s) of compounds 4a, 4b and 4c.**

| Samples | MIC | MQSIC | $IC_{50}$ |
|---|---|---|---|
| **4a** | 512 | 256 | 62.71±5.11 |
| **4b** | 256 | 64 | 28.56±1.24*** |
| **4c** | 256 | 128 | 107.45±8.56*** |
| **Vanillin** | 256 | 128 | 56.75±4.32 |

Statistical analysis was performed with Sidak's multiple comparisons test using two-way ANOVA; ***$p < 0.001$.

of violacein formation at 64 µg/mL). These results demonstrate that compounds **4a**, **4b** and **4c** possess anti-quorum sensing activity.

**Determination of the minimum inhibitory concentration (MIC) and minimum quorum sensing inhibitory concentration (MQSIC).** Table 4 summarizes the minimum inhibitory concentration (MIC) and minimum quorum sensing inhibitory concentration (MQSIC) of compounds **4a**, **4b** and **4c**, the most promising antibacterial tetrazine derivatives. Compounds **4a**, **4b** and **4c** revealed MIC and MQSIC values ranging from 256 to 512 µg/mL and from 64 and 256 µg/mL, respectively (Table 4). The concentrations of **4a**, **4b** and **4c** that inhibited 50% of violacein production were found to be 62.71, 28.56 and 107.45 µg/mL, respectively, vs vanillin ($IC_{50}$: 56.75 µg/mL).

## Discussion

Combating antimicrobial resistance (AMR) has been declared as a priority concern by the World Health Organization (WHO) ever since antibacterial AMR was directly responsible for 1.27 million global deaths in 2019 [4]. Adequate research and development on quality diagnosis, effective treatment of bacterial infections, and innovation are among the strategies that can slow down and eliminate bacterial drug resistance. To contribute toward the identification of effective treatments against drug resistant bacterial infections, numerous scientists have investigated the inhibitory effects of natural and synthetic compounds against different pathogens. This observation is exemplified by the number of recently published review articles on the antibacterial activity of heterocyclic compounds, including benzothiazole derivatives [39–41], tetrazine compounds [42,43], among others. On the other hand, the implication of biofilm formation [6,44,45] and quorum sensing system [46–48] in developing bacterial drug-resistance is prominent [49,50]. Thus, the search for effective antibacterial treatments with potential inhibition of biofilm formation and quorum sensing activity is valuable and might significantly contribute to antimicrobial drug discovery against multi-resistant bacteria. In point of fact, a number of antibacterial chemotypes were reported to exhibit anti-quorum sensing [51,52] and anti-biofilm activity [53,54]. However, the identification of antibacterial pharmacophores with novel features is still needed since none of these promising hit compounds is indicated to have entered the clinical trial phase, perhaps the reason being their unfavourable pharmacokinetic characteristics, their safety limits, and so on. As the benzothiazole and tetrazine rings hold promise as antibacterial potential candidates and that previously prepared derivatives from compounds bearing both the pharmacophores revealed non cytotoxicity against normal and cancer cells, the present study evaluated the antibacterial efficacy of 1,2,3,5-tetrazine amalgamated benzothiazole derivatives.

As a result, the synthesized 1,2,3,5-tetrazine derivatives showed MIC and MBC values ranging from 8 to 256 µg/mL and from 64 to 256 µg/mL, respectively, thus highlighting the antibacterial activity of these compounds. Compound **4b** was found to be the most potent, followed by compound **4c** and **4a**. Noteworthy, compound **4b** harbours more benzothiazole rings than its counterparts **4a** and **4c**, thus justifying the high inhibition of selected bacteria by compound **4b**.

Similarly, the compound **4a** contain a number of nitro groups, which might have aided in the observed antibacterial activity. By contrast, compound **4c** bears only two nitro groups, whereas compound **4a** do not contain any nitro moiety.

The chemical and physical properties of the nitro group ($-NO_2$) including its size, electron-withdrawing ability, polarity, ability to form hydrogen bonds and redox properties contribute to its key role in the action of many drugs, especially anti-microbial agents [55]. Accumulated evidence has shown the significant role of the nitro group in the inhibition of several bacteria [56–58].

The anti-biofilm and anti-quorum sensing effects of the most promising derivatives were also evaluated. As a result, compounds **4a**, **4b** and **4c**, inhibited biofilm formation with minimum biofilm inhibitory concentrations (MBIC$_{50}$s) ranging from 5.29 to 54.91 μg/mL, 7.79 to 45.23 μg/mL, and 8.56 to 87.35 μg/mL for compounds **4a**, **4b** and **4c**, respectively. Moreover, compounds **4a**, **4b** and **4c** displayed minimum biofilm eradication concentrations (MBIC$_{50}$s) of 98.62 and 60.98 μg/mL, 54.21 and 15.70 μg/mL, as well as 105.08 and 29.55 μg/mL, respectively, when tested against *Klebsiella aerogenes* and *Acinetobacter baumannii.* As shown in the supporting information (S1 and S2 Fig), the sub-inhibitory concentrations of compounds **4a**, **4b** and **4c** did not significantly affect the bacterial growth, biofilm formation and bio-logical synthesis of violacein over the tested time period, thus justifying the selective inhibitory effects of the test com-pounds on these specific bacterial processes rather than a general decrease in the bacterial cell viability. These results are consistent with previously reported data on the anti-quorum sensing effects of benzothiazole derivatives containing an isopropanolamine moiety [51], pyrazole-based benzothiazoles [59], etc. Moreover, series of 2-azidobenzothiazoles [51] and benzothiazole–urea hybrids [60] were reported to exhibit anti-biofilm activity. On the other hand, tetrazine groups are reputed for their antibacterial potential [61,62]. In general, the antibacterial activity of benzothiazole deriva-tives has been attributed to binding onto enzymes that are important for essential processes in the bacterial cells, such as cell-wall synthesis, cell division, and DNA replication [63,64]. Inhibition of violacein formation has been used as an approach to determine anti-quorum sensing activity [65]. The antibacterial, anti-biofilm, and anti-quorum sensing (QS) activities of the synthesized 1,2,3,5-tetrazine-benzothiazole hybrids (**4a-4c**) can be attributed to their unique structural features and potential interactions with bacterial targets. The potent bactericidal effects of **4b** (MIC: 8–64 μg/mL) likely stem from the benzothiazole moieties. These heterocycles may inhibit DNA gyrase or topoisomerase IV, which is a critical enzyme for bacterial DNA replication [63]. The additional benzothiazole rings in **4b** could improve target binding affinity compared to **4a** and **4c**. At sub-MIC concentrations, **4a-4c** selectively inhibited biofilm formation and violacein production without affecting bacterial growth. These results suggest the disruption of extracellular polymeric substances (EPS), since the azo and tetrazine groups are well known to interfere with EPS synthesis [66]. Growing evidence has shown that QS interference by violacein suppression might result from the inhibition of the CviR/I system in *C. viola-ceum* [65,67]. The planar structure of compound **4b** may competitively bind LuxR-type receptors, thus blocking the signal transduction [51].

## Conclusions

This work has unveiled the antibacterial, anti-quorum sensing and anti-biofilm formation of selected 1,2,3,5-tetrazine-benzothiazole hybrids, which can be prospected as potential pharmacophores for the discovery of effective antibacterial agents. Future studies could dissect the mechanisms of action of the antibacterial compounds using transcriptional analy-sis such as RT-qPCR of QS genes, or genetic knockouts to confirm target specificity. Nevertheless, our findings contribute to the rationale for developing sub-MIC therapies targeting virulence, particularly in biofilm-associated infections where conventional antibiotics have failed due to several factors, including the physical barrier of the biofilm matrix, slow bacte-rial growth rates, and the presence of persistent cells within the biofilm.

## Supporting information

**S1 Fig. Effect of selected compounds at sub-inhibitory concentrations on biofilm and bacterial growth inhibition.** **A:** effect of compound **4a**, **B:** effect of compound **4b**, **C:** effect of compound **4c**, **MIC:** minimum inhibitory concentration. (DOCX)

**S2 Fig. Effect of selected compounds 4a, 4b and4c on*Chromobacterium violaceum* growth at sub-inhibitory concentrations. MIC:** minimum inhibitory concentration.
(DOCX)

## Author contributions

**Conceptualization:** Jean Paul Dzoyem, Joseph Tsemeugne, Boniface Pone Kamdem, Fabrice Fekam Boyom, Lyndy Joy McGaw.

**Data curation:** Boris Arnaud Kuate.

**Formal analysis:** Joseph Tsemeugne, Rostand Foyou Meupiap, Boris Arnaud Kuate, Pierre Mkounga.

**Funding acquisition:** Jean Paul Dzoyem, Joseph Tsemeugne, Fabrice Fekam Boyom, Lyndy Joy McGaw.

**Investigation:** Rostand Foyou Meupiap, Boris Arnaud Kuate, Pierre Mkounga.

**Methodology:** Joseph Tsemeugne, Rostand Foyou Meupiap, Boris Arnaud Kuate.

**Project administration:** Jean Paul Dzoyem, Joseph Tsemeugne, Boniface Pone Kamdem, Fabrice Fekam Boyom, Lyndy Joy McGaw.

**Resources:** Jean Paul Dzoyem, Joseph Tsemeugne, Pierre Mkounga, Fabrice Fekam Boyom, Lyndy Joy McGaw.

**Software:** Rostand Foyou Meupiap, Boris Arnaud Kuate, Pierre Mkounga.

**Supervision:** Jean Paul Dzoyem, Boniface Pone Kamdem, Fabrice Fekam Boyom.

**Validation:** Joseph Tsemeugne, Boniface Pone Kamdem, Fabrice Fekam Boyom, Lyndy Joy McGaw.

**Visualization:** Jean Paul Dzoyem, Boniface Pone Kamdem, Pierre Mkounga, Fabrice Fekam Boyom, Lyndy Joy McGaw.

**Writing – original draft:** Rostand Foyou Meupiap, Boris Arnaud Kuate.

**Writing – review & editing:** Joseph Tsemeugne, Boniface Pone Kamdem, Pierre Mkounga, Lyndy Joy McGaw.

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
