## [Decision Letter · Decision Letter 0]

28 Mar 2025

PONE-D-25-01821Antibacterial, antibiofilm and anti-quorum sensing activities of 1,2,3,5-tetrazine derivatives linked to a benzothiazole moietyPLOS ONE

Dear Dr. Pone Kamdem,

Thank you for submitting your manuscript to PLOS ONE. After careful consideration, we feel that it has merit but does not fully meet PLOS ONE’s publication criteria as it currently stands. Therefore, we invite you to submit a revised version of the manuscript that addresses the points raised during the review process.

The submitted manuscript has scientific merit and it is positively valued to work with chemically synthesized compounds by the authors' own team in the context of bacterial quorum sensing inhibition. However, in light of the review, the manuscript must correct and clarify aspects related to the methodology, specifically the relationship between MICs, MIBCs, and the biomass necessary to perform QS and form biofilms. Additionally, the authors should delve into the chemical and molecular mechanisms that explain the observed results. This should be done by proposing chemical or molecular biology assays that can elucidate the pathways through which the derived triazines exert their action.

We look forward to receiving your revised manuscript.

Kind regards,

M. Alejandro Dinamarca, Dr.

Academic Editor

PLOS ONE

Journal Requirements:

Additional Editor Comments:

The article titled Antibacterial, antibiofilm and anti-quorum sensing activities of 1,2,3,5-tetrazine derivatives linked to a benzothiazole moiety, is a work that evaluates a series of organic compounds derived from chemical synthesis conducted by the group of authors. Specifically, the authors evaluate their synthesis products in two related bacterial behavior systems, which are quorum sensing communication and biofilm formation. The evaluations are conducted on a series of bacterial pathogens or pathogenicity models such as: Klebsiella aerogenes ATCC 130148, Acinetobacter baumannii ATCC BAA-1605, Enterococcus faecium ATCC700221, Staphylococcus epidermidis. Additionally, the ability to inhibit the quorum sensing mechanism is evaluated using the production of violacein in Chromobacterium violaceum ATCC12472 as a model. To methodologically relate their work, the authors propose an experimental design based on determining the respective MICs and MIBCs of each compound in concentrations of µg/ml. The results of the work are expressed in percentage values through known formulas.

The results are consistent with the experimental design; however, they present a methodological question that must be clarified by the authors. Specifically, if bacteria need a specific concentration to perform QS communication, how does this relate to the concentrations related to MICs and MIBCs used for each evaluated triazine? The same is considered for the formation or eradication of biofilms. This must be adequately clarified in the methodology section and in the results, with a special mention in the discussion section.

Finally, although the article is coherent and it is positively valued to work with compounds derived from synthesis, it is required to delve deeper methodologically or in the discussion regarding the molecular and chemical mechanisms that may be involved in the observed results.

The manuscript must be presented in the format required by PLOS.

Reviewers' comments:

Reviewer's Responses to Questions

**Comments to the Author**

1. Is the manuscript technically sound, and do the data support the conclusions?

Reviewer #1: Partly

2. Has the statistical analysis been performed appropriately and rigorously? 

Reviewer #1: N/A

3. Have the authors made all data underlying the findings in their manuscript fully available?

Reviewer #1: Yes

4. Is the manuscript presented in an intelligible fashion and written in standard English?

Reviewer #1: Yes

5. Review Comments to the Author

Reviewer #1: The manuscript addresses a relevant topic and evaluates new potential antimicrobial and antivirulence compounds. While the evaluated compounds show potential against relevant pathogens, there are some aspects of the microbiological assays that are not included and without which a solid conclusion can not be generated.

Major comments

There are some major methodological issues regarding the evaluation of biofilm inhibition and violacein production. While authors test different concentrations of the selected compounds below MIC and MBC, they do not take into consideration the potential effect of subinhibitory concentrations of these on bacterial growth. Therefore, it is not possible to determine if the lower biofilm formation and violacein production is due to the selective effect of the compounds on these cellular processes or to a reduction of bacterial cells. Normalization by total cell number (CFU or Optical density at 600 nm) or growth curves at the selected concentrations of the compounds should be included to support the conclusions.

Minor comments:

Introduction, L5: change "The main mechanisms of ..... includes" for "include" (plural)

Introduction, L7: "degradation" may be replace bt a more accurate concept as "structural modification"

Methods, 2.2.1: Please indicate why C. violaceum was cultured at 30 ºC instead of 26 ºC (its optimal growth temperature)

Methods 2.2.3: Delete ")" after gentamicin.

Please specify gentamicin concentrations used as control condition.

"The MBC was determined as the ...that completely inhibited the growth of bacteria". Because authors are evaluating bactericidal effect, the more accurate concept should be "that inhibited bacterial viability".

Methods, 2.3.1. Please specify vanillin concentration

Results, 3.1.2: Authors should include ATCC numbers for all strains (or none).

Results, 3.4.1, Figure 2. According to the figure and its Y-axis, lower concentrations of the selected compouds produced higher ihnibition of violacein production. Please check this information and plot´s axis names.

Discussion "By contrast, compound 4c bears only two nitro groups, whereas

compound 4c do not contain any nitro moiety". Authors mention compound 4c twice, please check.

Conclusion. Authors do not expose the conclusion of their research. Instead, in this section, they summarize the relevance and of their results. Please, include the conclusions.

6. PLOS authors have the option to publish the peer review history of their article (what does this mean? ). If published, this will include your full peer review and any attached files.

**Do you want your identity to be public for this peer review?** For information about this choice, including consent withdrawal, please see our Privacy Policy .

Reviewer #1: No

---

## [Author Response · Author response to Decision Letter 0]

25 Apr 2025

Manuscript number: [PONE-D-25-01821] - [EMID:c96f77a474010caa]

Title: Antibacterial, antibiofilm and anti-quorum sensing activities of 1,2,3,5-tetrazine derivatives linked to a benzothiazole moiety

Answers to Reviewers and Editor Comments

Additional Editor Comments:

The article titled Antibacterial, antibiofilm and anti-quorum sensing activities of 1,2,3,5-tetrazine derivatives linked to a benzothiazole moiety, is a work that evaluates a series of organic compounds derived from chemical synthesis conducted by the group of authors. Specifically, the authors evaluate their synthesis products in two related bacterial behavior systems, which are quorum sensing communication and biofilm formation. The evaluations are conducted on a series of bacterial pathogens or pathogenicity models such as: Klebsiella aerogenes ATCC 130148, Acinetobacter baumannii ATCC BAA-1605, Enterococcus faecium ATCC700221, Staphylococcus epidermidis. Additionally, the ability to inhibit the quorum sensing mechanism is evaluated using the production of violacein in Chromobacterium violaceum ATCC12472 as a model. To methodologically relate their work, the authors propose an experimental design based on determining the respective MICs and MIBCs of each compound in concentrations of µg/ml. The results of the work are expressed in percentage values through known formulas.

The results are consistent with the experimental design; however, they present a methodological question that must be clarified by the authors. Specifically, if bacteria need a specific concentration to perform QS communication, how does this relate to the concentrations related to MICs and MIBCs used for each evaluated triazine? The same is considered for the formation or eradication of biofilms. This must be adequately clarified in the methodology section and in the results, with a special mention in the discussion section.

Finally, although the article is coherent and it is positively valued to work with compounds derived from synthesis, it is required to delve deeper methodologically or in the discussion regarding the molecular and chemical mechanisms that may be involved in the observed results.

Response: We sincerely appreciate the Editor’s thoughtful comments and suggestions for improving our manuscript. We have now clarified in the methodology section that the sub-inhibitory concentrations tested (1/2x, 1/4x, 1/8x, 1/16x, and 1/32x MIC) were specifically selected to evaluate bacterial growth and biofilm formation while avoiding growth effects. We have also expanded the discussion to include potential molecular targets of the tested compounds.

The manuscript must be presented in the format required by PLOS.

Response: We have now presented the manuscript in the format required by PLOS. The changes have been colored red with yellow stripes across the manuscript.

Reviewers' comments:

Comments to the Author

Reviewer #1:

Major comments

There are some major methodological issues regarding the evaluation of biofilm inhibition and violacein production. While authors test different concentrations of the selected compounds below MIC and MBC, they do not take into consideration the potential effect of subinhibitory concentrations of these on bacterial growth. Therefore, it is not possible to determine if the lower biofilm formation and violacein production is due to the selective effect of the compounds on these cellular processes or to a reduction of bacterial cells. Normalization by total cell number (CFU or Optical density at 600 nm) or growth curves at the selected concentrations of the compounds should be included to support the conclusions.

Response: We appreciate the reviewer’s insightful comment regarding the potential effect of subinhibitory concentrations of the tested compounds on bacterial growth. To clarify this point, we provided data of the kinetics of bacterial growth and biofilm formation in response to subinhibitory concentrations of the tested compounds, alongside our biofilm and violacein production experiments. We have now explicitly discussed this point in the revised manuscript to ensure clarity for readers. The changes have been colored red with yellow stripes in the main text. Thank you for raising this important consideration.

Minor comments:

Introduction, L5: change "The main mechanisms of ..... includes" for "include" (plural)

Introduction, L7: "degradation" may be replace bt a more accurate concept as "structural modification"

Response: The appropriate correction has been made.

Methods, 2.2.1: Please indicate why C. violaceum was cultured at 30 ºC instead of 26 ºC (its optimal growth temperature).

Response: The mistake has been corrected.

Methods 2.2.3: Delete ")" after gentamicin.

Please specify gentamicin concentrations used as control condition.

Response: The gentamicin range of concentrations used has been provided.

"The MBC was determined as the ...that completely inhibited the growth of bacteria". Because authors are evaluating bactericidal effect, the more accurate concept should be "that inhibited bacterial viability".

Response: The appropriate correction has been made.

Methods, 2.3.1. Please specify vanillin concentration

Response: The appropriate correction has been made.

Results, 3.1.2: Authors should include ATCC numbers for all strains (or none).

Response: ATCC numbers are now provided only once, in the methodology section.

Results, 3.4.1, Figure 2. According to the figure and its Y-axis, lower concentrations of the selected compouds produced higher ihnibition of violacein production. Please check this information and plot´s axis names.

Response: We appreciate the reviewer’s comments and suggestions for improving our manuscript. The mistake has been corrected. Y-axis of Figure 2 is the % of violacein production, where higher concentrations of the selected compounds produced lower violacein production.

Discussion "By contrast, compound 4c bears only two nitro groups, whereas compound 4c do not contain any nitro moiety". Authors mention compound 4c twice, please check.

Response: We agree with the reviewer’s inquisitiveness. The change has now been amended as suggested.

Conclusion. Authors do not expose the conclusion of their research. Instead, in this section, they summarize the relevance and of their results. Please, include the conclusions.

Response: The conclusion section has now been re-written as suggested.

We request the Editorial office for re-evaluation and kind consideration.

---

## [Editor Report · Decision Letter 1]

6 May 2025

Antibacterial, antibiofilm and anti-quorum sensing activities of 1,2,3,5-tetrazine derivatives linked to a benzothiazole moiety

PONE-D-25-01821R1

Dear 

Dr. Boniface Pone Kamdem,

We’re pleased to inform you that your manuscript has been judged scientifically suitable for publication and will be formally accepted for publication once it meets all outstanding technical requirements.

Kind regards,

M. Alejandro Dinamarca, Dr.

Academic Editor

PLOS ONE
---

## [Editor Report · Acceptance letter]

PONE-D-25-01821R1

PLOS ONE

Dear Dr. Pone Kamdem,

I'm pleased to inform you that your manuscript has been deemed suitable for publication in PLOS ONE. Congratulations! Your manuscript is now being handed over to our production team.

Kind regards,

on behalf of

Mr M. Alejandro Dinamarca

Academic Editor

PLOS ONE